# Effect of Bacterial Amyloid Protein Phenol−Soluble Modulin Alpha 3 on the Aggregation of Amyloid Beta Protein Associated with Alzheimer’s Disease

**DOI:** 10.3390/biomimetics8060459

**Published:** 2023-10-01

**Authors:** Bushu Peng, Shaoying Xu, Yue Liang, Xiaoyan Dong, Yan Sun

**Affiliations:** Department of Biochemical Engineering, School of Chemical Engineering and Technology, Key Laboratory of Systems Bioengineering and Frontiers Science Center for Synthetic Biology (Ministry of Education), Tianjin University, Tianjin 300350, China; shuzi0410@tju.edu.cn (B.P.); tinaxu@tju.edu.cn (S.X.); liangyue@tju.edu.cn (Y.L.)

**Keywords:** Alzheimer’s disease, phenol soluble modulins, amyloid β−protein, molecular interactions, molecular dynamics simulation

## Abstract

Since the proposal of the brainstem axis theory, increasing research attention has been paid to the interactions between bacterial amyloids produced by intestinal flora and the amyloid β−protein (Aβ) related to Alzheimer’s disease (AD), and it has been considered as the possible cause of AD. Therefore, phenol−soluble modulin (PSM) α3, the most virulent protein secreted by *Staphylococcus aureus*, has attracted much attention. In this work, the effect of PSMα3 with a unique cross−α fibril architecture on the aggregation of pathogenic Aβ_40_ of AD was studied by extensive biophysical characterizations. The results proposed that the PSMα3 monomer inhibited the aggregation of Aβ_40_ in a concentration−dependent manner and changed the aggregation pathway to form granular aggregates. However, PSMα3 oligomers promoted the generation of the β−sheet structure, thus shortening the lag phase of Aβ_40_ aggregation. Moreover, the higher the cross−α content of PSMα3, the stronger the effect of the promotion, indicating that the cross−α structure of PSMα3 plays a crucial role in the aggregation of Aβ_40_. Further molecular dynamics (MD) simulations have shown that the Met1−Gly20 region in the PSMα3 monomer can be combined with the Asp1−Ala2 and His13−Val36 regions in the Aβ_40_ monomer by hydrophobic and electrostatic interactions, which prevents the conformational conversion of Aβ_40_ from the α−helix to β−sheet structure. By contrast, PSMα3 oligomers mainly combined with the central hydrophobic core (CHC) and the C−terminal region of the Aβ_40_ monomer by weak H−bonding and hydrophobic interactions, which could not inhibit the transition to the β−sheet structure in the aggregation pathway. Thus, the research has unraveled molecular interactions between Aβ_40_ and PSMα3 of different structures and provided a deeper understanding of the complex interactions between bacterial amyloids and AD−related pathogenic Aβ.

## 1. Introduction

Alzheimer’s disease (AD), one of the most common neurodegenerative diseases, is widely recognized as the major cause of dementia. The pathological feature of AD is the plaques formed between neurons in the brain by abnormal levels of the amyloid β−protein (Aβ) and the intracellular neurofibrillary tangles caused by the hyperphosphorylation of tau [1,2,3]. The amyloids cascade hypothesis suggests that Aβ deposition triggers neuronal dysfunction and death in the brain associated with the disease [4]. In recent years, increasing evidence has identified that the gastrointestinal tract and the central nervous system can be bi−directionally linked through the microbiota–gut–brain axis, indicating potentially common pathogenic mechanisms among these diseases. Thus, the intestinal microbiota not only modulates the gastrointestinal tract, but also influences the function and development of the brain. It has been discovered that gut dysbiosis is linked to neurodegenerative diseases, such as AD, depression, and Parkinson’s disease [5,6,7,8,9,10,11]. Studies have shown that the interaction of bacterial amyloids produced by intestinal flora and pathogenic amyloids related to neurodegenerative diseases may be one of the main triggers of AD [12,13]. Recent studies on bacterial amyloids have shown that CsgA, the major subunit of bacterial amyloids curli secreted by *Escherichia coli*, could accelerate the formation of Aβ_40_ fibrils in vitro [14]. The FapC amyloid fragment has been shown to accelerate Aβ aggregation, inducing the neurotoxicity and pathological features of AD [15]. These results demonstrate that bacterial amyloids can interact with pathogenic amyloids, potentially influencing the development of neurodegenerative diseases through a novel mechanism of amyloidosis. However, the specific mechanisms behind the interaction between bacterial amyloids and Aβ remain elusive and require further exploration.

Phenol−soluble modulins (PSMs) are amphiphilic peptides with a wide range of cytolytic activities, and they serve as an important virulence factor secreted by *Staphylococcus aureus* (*S. aureus*), a representative member of the intestinal flora. The PSM peptides play an important role in stabilizing the biofilm structure by enhancing the biofilm matrix, resisting mechanical stress, and degrading enzyme catabolism [16]. PSMα3 is the most cytotoxic and soluble of the PSM peptides [17] and has attracted much attention because of its cross−α fibril structure [13]. The unique conformation of PSMα3 potentiates its toxicity to human cells [18]. The cross−α structure is a newly discovered self−assembly pattern that differs from the typical β−sheet structure. It exhibits an accumulation of α−helix structures perpendicular to the fibril axis (Appendix A) [19]. Previous studies have shown that PSM peptides can selectively cross−seed each other, indicating complex interactions between different PSM peptides [20]. A model has been proposed to describe the complex interaction among various PSM peptides in biofilm formation. The rapidly aggregated PSMα3 accelerates the assembly of PSMα1, and PSMα1 accelerates the aggregation of the remaining PSM peptides [20]. PSM monomers can aggregate into bacterial functional amyloids and cross−seed each other, accelerating the aggregation process. However, the effect of PSMα3 produced by *S. aureus* with a unique cross−α structure on the aggregation of Aβ remains poorly understood.

Therefore, the effect of PSMα3 on the aggregation process of Aβ_40_ was investigated in this study, using various experimental methods such as Thioflavin T (ThT) fluorescence assays, circular dichroism (CD) experiments, and atomic force microscopy (AFM). Moreover, the interaction mechanism between PSMα3 and Aβ_40_ was analyzed via molecular dynamics (MD) simulations to reveal the influencing mechanism of the intestinal flora on AD at the molecular level.

## 2. Materials and Methods

### 2.1. Materials

Aβ_40_ (>95%) was provided by GL Biochem (Shanghai, China). PSMα3 (>95%) was provided by ZiYu Biotech (Shanghai, China). 1,1,1,3,3,3−Hexafluoro−2−propanol (HFIP) and ThT were provided by Sigma (St. Louis, MO, USA). Antibodies 6E10 and OC were obtained from Covance (Dedham, MA, USA). Secondary antibodies anti−mouse IgG and anti−rabbit IgG were provided by Beyotime (Shanghai, China).

### 2.2. Preparation of Aβ_40_ and PSMα3 Monomers

Aβ_40_ was pretreated according to the literature [21]. Aβ_40_ was dissolved in HFIP at a concentration of 1.0 mg/mL and placed stably at 4 °C for 2 h. The solution was then ultrasonicated for 20 min in an ice bath to destroy the Aβ_40_ aggregates. Then, a freeze dryer (FreeZone, Labconco, Kansas City, MI, USA) was used to remove HFIP to obtain fluffy flocculent Aβ_40_, which was stored at −20 °C. PSMα3 was treated by the same procedure as Aβ_40_ and stored at −20 °C until the beginning of the experiment to ensure PSMα3 was in a monomeric conformation [21]. For use, Aβ_40_ and PSMα3 monomers were dissolved in NaOH (20 mM) and ultrasonicated for 5 min, then centrifugated at 16,000× *g* for 20 min at 4 °C. Then, the peptides were diluted to a final concentration of 25 μM with HEPES buffer (30 mM, pH 7.4).

### 2.3. Preparation of PSMα3 Seeds

The PSMα3 monomer was incubated in an air−bath shaker at 130 rpm and 37 °C for different times (1 to 24 h) and then centrifugated at 16,000× *g* for 20 min at 4 °C. PSMα3 seed suspension was obtained by adding PSMα3 solution into HEPES buffer at a concentration of 5 μM.

### 2.4. ThT Fluorescence Assay

A 200 μL sample containing Aβ_40_, PSMα3, and equimolar ThT was mixed evenly and added to 96−well plates at a concentration of 25 μM for both Aβ_40_ and ThT, and the concentration of PSMα3 seeds was 5 μM. The samples were incubated in a fluorescence plate reader (PerkinElmer, LS55, Waltham, MA, USA) with shaking at 30 min intervals at 37 °C. The ThT emission fluorescence at 480 nm was measured by excitation wavelengths at 440 nm. At least three parallel groups were set up for each experiment and the results were averaged after subtracting the corresponding background.

### 2.5. AFM Experiments

The Aβ_40_ monomer (25 μM) solution was cultured with or without the PSMα3 monomer and PSMα3 seeds at 37 °C for 100 h. The 50 μL samples were dropped on a flat mica sheet, and then other impurities were carefully washed away with ultrapure water after 10 min. The morphological characteristics of the samples were observed using atomic force microscopy (CSPM5500, Benyuan, Guangzhou, China). The pixels of the images were set to 1024 × 1024, and at least three separate areas of each sample were scanned by AFM to ensure consistency.

### 2.6. CD Experiments

The secondary structure of Aβ_40_ aggregates (25 μM) cultured with or without PSMα3 was examined by a CD spectrometer (J−810, JASCO, Tokyo, Japan). The samples were incubated in an air bath shaker for 100 h and then placed in a quartz cell with a 1 mm optical range. The CD spectra were scanned between 260 nm and 190 nm continuously at 100 nm/min with a bandwidth of 1 nm. The CD spectra were averaged from three parallel samples after subtracting the background.

### 2.7. Dot−Blot Assays

Ten microliters of Aβ_40_ (25 μM) co−cultured with or without the PSMα3 monomer and seeds was spotted on a nitrocellulose membrane and dried at room temperature for 1 h. The membrane was blocked with 10% skimmed milk for 1 h. After washing three times with TBS−T buffer (20 mM Tris−HCl, 150 mM NaCl, and 0.05% Tween 20, pH 7.4), the membranes were incubated with the Aβ sequence−specific 6E10 antibody (1:1000) and Aβ fibril−specific OC antibody (1:1000) for 1 h at room temperature and washed three times with TBS−T buffer. The membranes were then incubated with horseradish peroxidase−labeled anti−mouse IgG (1:2000) or anti−rabbit IgG (1:2000) for 1 h at room temperature and washed three times with TBS−T buffer. Finally, color development was performed with an ECL chemiluminescence kit (Beyotime, Shanghai, China). Images were taken using a chemiluminescence imaging system (SH−Compact 523, Shenhua, Hangzhou, China).

### 2.8. MD Simulations

The structures of Aβ_40_ (PBD ID: 1BA4) [22] and PSMα3 (PBD ID: 5I55) [19] were selected as the templates. The initial structure of the PSMα3 monomer was taken from a PSMα3 fibrous structure formed by a set of twenty helices (PBD ID: 5I55) [19], and one of the helices was selected as the template. To construct a PSMα3 oligomer model, three PSMα3 monomers were placed in parallel with a minimum distance of 1.0 nm to construct an isolated trimeric system [23]. The complex system was formed by docking Aβ_40_ and PSMα3 using AutoDock Vina [24]. All systems were simulated using the GROMACS 5.1.4 software package [25] with the AMBER 99SB force field [26]. The periodic simulation box was created and the minimum distance from the molecule to the edge of the box was set to 1 nm. The solvent water molecule TIP3P [27] was then filled in the system and Na^+^ and Cl^−^ ions were added to neutralize the electrostatic charges of the simulated systems. The systems were then optimized by energy minimization and performed a solvent equilibrium in 50,000 steps at a constant NVT of 100 ps and 300 K, followed by a pressure equilibrium of 100 ps at NPT conditions. The MD simulations were then performed for 50 ns with a step size of 1 fs. Previous studies have shown that a simulation time of 50 ns is sufficient to bring the complex to equilibrium and to illustrate conformational transitions in peptides [28,29]. The free binding energy between Aβ_40_ and PSMα3 was calculated by g_mmpbsa [30]. Pymol was used to obtain snapshots. Three independent MD simulations were carried out for each system.

## 3. Results and Discussion

### 3.1. Effects of the PSMα3 Monomer on Aβ_40_ Aggregation

It has been reported that the cross−α structure of PSMα3 fibrils can specifically bind to the amyloid dye ThT and produce enhanced fluorescence [18]. Here, the aggregation kinetic of PSMα3 was measured by a ThT fluorescence assay. The aggregation of PSMα3 was observed as an immediate increase in ThT fluorescence intensity without a significant lag phase. The ThT fluorescence intensity peaked around 2.5 h and then gradually decreased until it reached equilibrium (Appendix A). The aggregation of PSMα3 was different from the standard sigmoidal kinetic behavior of Aβ_40_ [31,32], consistent with previous results [20,33]. Studies have shown that newly formed PSMα3 aggregates are unstable, but they can stabilize the structure by lateral association during aggregation [20]. The CD spectra showed a negative peak at 216 nm (typical β−sheet signal) for Aβ_40_ aggregates, indicative of a β−sheet−rich structure (Appendix A). In contrast, the CD spectra of PSMα3 aggregates showed a minimum at 222 nm, implying a reduced helicity of the α−helical structure [34]. Combined with the results of ThT fluorescence experiments in Appendix A, the above results suggested that PSMα3 formed the cross−α structure composed of an α−helix, which could bind to the amyloid dye ThT [18]. This is consistent with the results of X−ray diffraction spectroscopy and Fourier transform infrared spectroscopy [19,34].

The morphological characteristics of the Aβ_40_ and PSMα3 aggregates were examined using AFM. Aβ_40_ formed elongated fibrils after 72 h (Appendix A), consistent with previous results [35]. However, PSMα3 formed granular aggregates (Appendix A), different from the elongated fibril structures reported in the literature [19,34]. This was because differences in experimental conditions, such as peptide concentration, buffer ionic strength, pH, and shaking conditions, can affect the morphology of PSMα3 aggregates. Overall, these results demonstrate differences in structure and morphology between Aβ_40_ and PSMα3 aggregates.

To explore whether PSMα3 plays a role in Aβ_40_ aggregation, the kinetics of Aβ_40_ aggregation cultured with different concentrations of the PSMα3 monomer was monitored (Figure 1a,b). It can be seen that the maximum ThT fluorescence intensity of Aβ_40_ significantly decreased (Figure 1a), and the lag phase prolonged (Appendix A) with increasing the concentration of the PSMα3 monomer. The kinetic curves of 0 to 20 h in Figure 1a are magnified in Figure 1b, allowing a clearer observation of the short time changes in the ThT fluorescence intensity. Interestingly, when Aβ_40_: PSMα3 > 1:1, the aggregation of Aβ_40_ still followed the sigmoidal kinetic curve with reduced ThT fluorescence. The aggregation of Aβ_40_ appeared to be completely inhibited by the addition of equal concentrations of the PSMα3 monomer (purple line in Figure 1a). However, when Aβ_40_: PSMα3 < 1:1, the aggregation kinetics of Aβ_40_ no longer fitted the typical sigmoidal curve but tended to follow the aggregation kinetics of PSMα3 itself (green and blue lines in Figure 1a). This indicates that the ratio of Aβ_40_ to PSMα3 monomer influenced the type of aggregation. The amyloids with a higher amount than the other in the mixture dominated the aggregation dynamics and determined the amyloids aggregation pathway. To further explore the mechanism of inhibition, the relevant kinetic parameters, including *k*_n_*k*_+_ (related to the primary pathway) and *k*_2_*k*_+_ (related to the secondary pathway), were calculated using the Amylofit program [36]. The results revealed that both the *k*_n_*k*_+_ and *k*_2_*k*_+_ values decreased in the presence of the PSMα3 monomer, while the *k*_2_*k*_+_ value decreased more significantly (Appendix A). It indicated that the PSMα3 monomer mainly inhibited the secondary pathway of unseeded Aβ_40_ aggregation.

The morphology of Aβ_40_ incubated with different concentrations of the PSMα3 monomer is shown in Figure 1c. When Aβ_40_: PSMα3 > 1:1, the total density of fibrils detected by AFM decreased and gradually transformed into sparse short fibrils or some granular aggregates. When Aβ_40_: PSMα3 < 1:1, no fibrous structures can be observed (Figure 1c). This is consistent with the ThT fluorescence assay (Figure 1a), indicating that the PSMα3 monomer inhibited the aggregation of Aβ_40_ in a concentration−dependent manner and could change the aggregation pathway of Aβ_40_.

The inhibitory effect of the PSMα3 monomer on Aβ_40_ aggregation was further demonstrated by dot−blot assays with antibody 6E10 (Aβ_1–16_ sequence−specific) and antibody OC (fibril−specific) (Appendix A). The 6E10 antibody recognizes the Aβ_1–16_ sequence, and all of the Aβ species can be stained by it [37]. By staining with 6E10, there was no change in spot size or darkness in any Aβ or Aβ−PSMα3 species at 0 h and 100 h, indicating the stable presence of Aβ during incubation (Appendix A). At the beginning of incubation (0 h), Aβ fibrils were rarely detected by antibody OC in the presence or absence of the PSMα3 monomer (Appendix A, left). The OC−positive dots were detected in Aβ_40_ only after 100 h of incubation, indicating the formation of a large number of Aβ fibrils, consistent with the previous results [38]. However, after co−cultivation with the PSMα3 monomer, the positive response was significantly reduced by increasing the PSMα3 concentration (Appendix A, right), indicating that the PSMα3 monomer significantly inhibited the formation of Aβ_40_ fibrils, which was in agreement with the ThT and AFM results (Figure 1).

According to the above results, it can be concluded that the cross−interaction between the PSMα3 monomer and the Aβ_40_ monomer resulted in the formation of non−fibrillar heterogeneous aggregates (Figure 1c), similar to the hetero−oligomers formed by Aβ and human islet amyloid polypeptide (hIAPP) [39,40] or tau [41]. These heterogeneous structures interfered with the conformational transitions of Aβ_40_, resulting in reduced ThT fluorescence (Figure 1a). Moreover, Aβ_40_ and PSMα3 monomers were reported to retain α−helix and disordered structures during the initial stages of aggregation [19,22]. The diversity and instability of structures hindered their common nucleation process, resulting in difficult cross−seeding with each other, and thus leading to an increased lag phase [42].

### 3.2. Effect of PSMα3 Seeds on Aβ_40_ Aggregation

The secondary nucleation mechanism of Aβ suggests that preformed fibril seeds of Aβ could accelerate its aggregation process [43]. In addition, several different disease−associated misfolded proteins, such as hIAPP [44], tau [45], and α−synuclein [46], could also hybridize with Aβ and promote amyloids formation. Studies have shown that the main molecular mechanism for the formation of PSM peptides was secondary nucleation with selective cross−seeding ability [20]. To investigate whether PSMα3 seeds accelerate Aβ aggregation through cross−seeding, thereby aggravating AD, cross−seeding experiments were performed using 20% preformed PSMα3 seeds added to the Aβ_40_ monomer. It was observed that the aggregation of Aβ_40_ incubated with PSMα3 seeds fitted the sigmoidal kinetic curve with no significant change in the maximum ThT fluorescence intensity (Figure 2a). The results indicated that the PSMα3 seeds had weak effects on the aggregation pathway and the formation of the β−sheet structure of Aβ_40_. However, compared to the non−seeded aggregation, the lag phase of Aβ_40_ incubated with PSMα3 seeds significantly decreased (Appendix A). The aggregation of Aβ_40_ incubated with PSMα3 seeds cultured for 2.5 h showed the shortest lag phase (~33.8 h), indicating that the PSMα3 seeds cultured for 2.5 h showed the most significant promotion effect on Aβ_40_ aggregation. This might be because the ThT fluorescence intensity peaked at 2.5 h in the aggregation kinetic curve of PSMα3 (Appendix A), indicating that PSMα3 contained the highest content of the cross−α structure after 2.5 h of aggregation. A kinetic analysis revealed a significant increase in *k*_n_*k*_+_ and a slight decrease in *k*_2_*k*_+_ in the presence of the PSMα3 seeds (Appendix A). Similarly, the PSMα3 seeds at 2.5 h showed the highest *k*_n_*k*_+_ value, in agreement with the lag phase results (Appendix A). These results confirmed that the PSMα3 seeds decreased the lag phase of Aβ_40_ by promoting the primary pathway, which was different from the role of the PSMα3 monomer (Figure 1a). This may be due to the fact that the PSMα3 seeds were formed by the cross−α structures, which were highly similar to the β−sheet structures [19]. The PSMα3 seeds could serve as templates to facilitate cross−seeding and promote the formation of β−sheet structures, thus accelerating the aggregation of Aβ_40_ [47]. Meanwhile, the promotion effect of the PSMα3 seeds on Aβ_40_ aggregation was correlated with the content of the cross−α structure. The higher content of the cross−α structure, the more obvious the promotion effect of the PSMα3 seeds.

To investigate the effect of the PSMα3 seeds on the secondary structure of Aβ_40_, the CD spectra of Aβ_40_ induced by the PSMα3 seeds were recorded (Figure 2b). They showed a minimum at approximately 216 nm, and the peak intensity was close to that of Aβ_40_ cultured alone (Figure 2b). The contents of different secondary structures were calculated from the CD spectrum (Table 1), and it was observed that the content of α−helix structures slightly increased after co−culture with the PSMα3 seeds. This might be because the main secondary structure of the PSMα3 aggregates is α−helix [34] (Appendix A), thus increasing the content of the α−helix structure in the mixed system. In contrast, the changes in the content of the β−sheet structure were not obvious (~70%) (Table 1), indicating that the PSMα3 seeds had little effect on the content of the β−sheet structure in the Aβ_40_ aggregates. But compared with the PSMα3 seeds of other incubation times, the cross−seeding of Aβ_40_ and PSMα3 seeds at 2.5 h had the highest content of α−helix structures (29.3%) and the lowest content of β−sheet structures (63.6%). This also indicated that PSMα3 seeds cultured for 2.5 h contained the highest content of cross−α, consistent with the aggregation kinetics of PSMα3 (Appendix A). Their interaction with Aβ_40_ had the most significant impact, which affected the conformational transitions of Aβ_40_, increased the content of the α−helix structure, and decreased the content of the β−sheet structure. Further observation of the aggregate morphology showed that Aβ_40_ induced by PSMα3 seeds forms more fibrils than Aβ_40_ alone, indicating that the PSMα3 seeds promoted the formation of Aβ_40_ fibrils. Dot−blot assays also showed that Aβ_40_ co−cultured with PSMα3 seeds exhibited significant OC−positive dots unaffected by increasing the incubation time with the PSMα3 seeds, indicating the substantial formation of Aβ fibrils (Appendix A), consistent with the results of ThT and AFM experiments (Figure 2).

### 3.3. Molecular Docking of PSMα3 and Aβ_40_ Monomer

To analyze the PSMα3 monomer (PSMα3m) and oligomer (PSMα3o) binding to Aβ_40_ at the molecular level, the corresponding composite models were constructed by docking the PSMα3m (PBD ID: 5I55) [19] or the PSMα3o with the Aβ_40_ monomer (PBD ID: 1BA4) [22] using AutoDock Vina. The docking results demonstrated that the best binding conformation of the Aβ_40_−PSMα3m complex had a binding free energy of −6.2 kcal/mol in the top 10 binding conformations (Appendix A), similar to the binding energies of Aβ and other amyloids, such as Aβ and the hIAPP monomer (~−6.4 kcal/mol) [48,49]. In contrast, the best binding conformation of the Aβ_40_−PSMα3o complex had a binding free energy of −13.0 kcal/mol (Appendix A), over 2.1−fold higher than that of the Aβ_40_−PSMα3m complex, implying that the PSMα3o binds to the Aβ_40_ monomer to form a more stable complex than the PSMα3m. This might be because the Aβ_40_ monomer and the PSMα3m are mostly in α−helix and disordered conformations [36]; the structural disorder and dynamic instability led to difficulties in their binding. However, the PSMα3o formed a stable cross−α structure [23], which facilitated a more stable binding to the Aβ_40_ monomer.

### 3.4. MD Simulations on PSMα3 and Aβ_40_ Monomers and Their Complex

To investigate the effects of the PSMα3m on the conformational transitions of Aβ_40_, the optimal conformation of the Aβ_40_−PSMα3m complex was selected as the initial conformation, and 50 ns MD simulations were performed with Aβ_40_ or the PSMα3m alone as the control for comparison. The root mean square deviation (RMSD) was calculated first to assess the stability of the simulated systems. As shown in the RMSD plots (Figure 3a), the RMSD values of all three systems initially increased rapidly and then stabilized after 30 ns at around 0.4 nm, 0.8 nm, and 0.9 nm, respectively, indicating that all three systems reached equilibrium within 20 ns. It can be seen that the initial equilibrium speed of the Aβ_40_−PSMα3m complex was slower than that of Aβ_40_ alone, and its RMSD value was higher than that of Aβ_40_ alone. These results indicate that the PSMα3m affected the conformational changes of Aβ_40_ in the initial stage of the simulations and reduced system stability.

To identify the key residues during the interaction between the PSMα3m and the Aβ_40_ monomer, the root mean square fluctuation (RMSF) of residues in the last 20 ns of the MD simulations was calculated. It was found that the RMSF values of Aβ_40_ residues in the Aβ_40_−PSMα3m complex were generally smaller than that of Aβ_40_ alone (Figure 3b), indicating that the tight binding of the PSMα3m reduced the flexibility of residue fluctuations of Aβ_40_. Moreover, the fluctuations of the Asp1−His6 and His14−Val36 regions in Aβ_40_ significantly decreased by the presence of the PSMα3m, indicating that the PSMα3m exhibits a stronger impact on these regions. These results suggested that the PSMα3m tightly bound the Asp1−His6 and His14−Val36 regions in Aβ_40_ and exhibited a disruptive effect on the Asp23−Lys28 salt bridge in Aβ_40_. Hydrophobic interactions and the salt bridge have been reported to play a critical role in stabilizing the structure of Aβ fibrils [50]. The disruption of the salt bridge prevented the formation of the β−sheet structure, thus reducing the structural stability of Aβ_40_ [51].

Furthermore, other structural parameters of different systems were analyzed to estimate the fast conformational transitions of Aβ_40_. The results showed that the binding of the PSMα3m increased the radius of gyration (*R*_g_) and the solvent−accessible surface area (SASA) of Aβ_40_ (Appendix A), indicating that the PSMα3m changed the conformational denseness of Aβ_40_ and increased the exposure of the contact area to the solvent, thus destabilizing the system, consistent with the RMSD results (Figure 3a). In addition, the PSMα3m reduced the number of intramolecular hydrogen bonds (H−bonds) of Aβ_40_ (Appendix A), which was detrimental to maintaining the structural stability of Aβ_40_ [52].

To observe the effect of the PSMα3m on the structure of the Aβ_40_ monomer, snapshots of Aβ_40_ and the Aβ_40_−PSMα3m complex in the initial and equilibrium states were extracted (Figure 3c,d). The conformational transitions from the initial α−helix structure to a random coil structure were observed in the Aβ_40_ monomer, which is a critical step in forming a highly ordered β−sheet structure for Aβ aggregation [53]. However, the α−helix structure of Aβ_40_ in the Aβ_40_−PSMα3m complex was well preserved (Figure 3d), indicating that the PSMα3m inhibited the conformational transitions of the Aβ_40_ monomer.

The content of secondary structures in the equilibrium state showed that Aβ_40_ alone contained only 13.8% of the α−helix structure and 4.5% of the β−bridge structure (Appendix A), which was a transition structure that transformed into the β−sheet structure [54]. The results indicate that the Aβ_40_ monomer transformed from the α−helix to the coil structure, and then formed the β−sheet structure, which is consistent with previous studies [55,56]. However, the binding of the PSMα3m resulted in a higher content of the α−helix structure (36.3%) than that of Aβ_40_ alone, accompanied by a decrease in the content of coil (29.1%) and turn structures (24.1%), and the β−bridge structure was not formed (Appendix A). These results showed that the PSMα3m inhibited the conformational transitions of Aβ_40_ from the α−helix structure to coil and turn structures, thus suppressing the formation of β−sheet structures, which prolonged the lag phase of Aβ_40_ and inhibited the fibrillization of Aβ_40_ (Figure 1a and Appendix A).

To demonstrate how the PSMα3m inhibits the conformational transitions of Aβ_40_, we investigated the changes in secondary structure during the whole MD simulation. For Aβ_40_ alone, the α−helix structures in the Gln15−Ala30 region were rapidly transformed into the coil and turn structures after 20 ns, and the β−bridge was formed in the Phe4−Ser8 region after 40 ns (Appendix A), consistent with the previous findings [55,56]. In contrast, the Gln15−Asp23 regions maintained their initial α−helix structure during the 50 ns MD simulation in the Aβ_40_−PSMα3m system. The regions of Val12−Gln15, which had the coil structure in the initial state, transformed into α−helix after 10 ns (Appendix A). Moreover, no β−sheet or β−bridge structures were detected during the whole 50 ns MD simulations. It is known that the maintenance of the α−helix structure inhibits the aggregation of Aβ into toxic oligomers [57]. Therefore, the PSMα3m inhibited Aβ_40_ aggregation by stabilizing the formation of α−helix structures and suppressing the generation of β−sheet structures.

To further investigate the intermolecular interactions between the PSMα3m and the Aβ_40_ monomer, the binding free energy between the PSMα3m and the Aβ_40_ monomer was calculated. The simulated trajectory of the last 20 ns was collected and calculated by the molecular mechanic Poisson–Boltzmann surface area (MM−PBSA) method [30]. The binding free energy between the PSMα3m and the Aβ_40_ monomer was −270.7 kJ/mol (Appendix A), indicating a high affinity of the PSMα3m and the Aβ_40_ monomer [48,49]. An analysis of the contribution of each energy component showed that van der Waals hydrophobic energy (Δ*G*_vdw_ = −373.5 kJ/mol) and electrostatic energy (Δ*G*_elec_ = −963.9 kJ/mol) were contributing more to the binding than other energies (Appendix A), indicating that hydrophobic and electrostatic interactions play a critical role in the binding of the PSMα3m to the Aβ_40_ monomer. The free energy was decomposed to search for the key residues in the interaction between the PSMα3m and the Aβ_40_ monomer. As shown in Figure 4, Asp1 and Leu17 of Aβ_40_ and Met1, Glu2, Phe3, Leu7, Leu14, and Phe18 of the PSMα3m contributed greatly to the binding of the PSMα3m to the Aβ_40_ monomer (binding free energy < −15.0 kJ/mol), in which the van der Waals force and electrostatic interactions were the main contributors (Appendix A).

It is known that the aggregation of Aβ is mainly controlled by hydrophobic interactions and H−bonding [58]. Six H−bonds were observed between the Aβ_40_ monomer and the PSMα3m, which were mainly located at the N−terminal of Aβ_40_ (Appendix A). It has been reported that the flexibility of the N−terminal plays a significant role in the aggregation and toxicity of Aβ [59]. This implies that the PSMα3m could alter the aggregation tendency of Aβ_40_ by decreasing the flexibility of the N−terminal. Moreover, the Aβ_40_ monomer is negatively charged [22], while the PSMα3m is positively charged [19]. The opposite charges lead to strong electrostatic interactions between Aβ_40_ and the PSMα3m, which facilitated their stable binding. In addition, some hydrophobic residues (Met1, Phe3, Leu7, Leu14, and Phe18) of the PSMα3m could provide hydrophobic interactions with the Aβ_40_ monomer. Therefore, the hydrophobic and electrostatic interactions between the PSMα3m and the Aβ_40_ monomer promote the tight binding and inhibit the conformational transitions of the Aβ_40_ monomer.

### 3.5. MD Simulations on Interactions between PSMα3 Oligomer and Aβ_40_ Monomer

To examine the effect of the PSMα3o on the conformational transitions of the Aβ_40_ monomer, the interaction of the PSMα3o with the Aβ_40_ monomer was analyzed using MD simulations (Figure 5). The RMSD value of the PSMα3o−Aβ_40_ system was stabilized at around 0.8 nm after 40 ns, indicating that the system reached equilibrium thereafter (Figure 5a). Similar to the PSMα3m, the PSMα3o increased *R*_g_ and SASA values and decreased the number of intramolecular H−bonds of the Aβ_40_ monomer (Appendix A), making the whole system unstable, and the RMSD values increased (Figure 5a). The RMSF value of the N−terminal residues of Aβ_40_ increased (Figure 5b), indicating that the PSMα3o increases the N−terminal flexibility of Aβ_40_ to accelerate the conformational transitions of Aβ_40_ [59]. Moreover, the PSMα3o significantly decreased the RMSF value of the Tyr10−Val40 region in Aβ_40_ (Figure 5b), indicating that the PSMα3o mainly interacted with this region and suppressed the dynamic fluctuation of the residues.

Notably, the Aβ_40_−PSMα3o complex at the equilibrium state (50 ns) formed a characteristic turn−like structure in Aβ_40_ (Figure 5c), which was different from the Aβ_40_−PSMα3m system (Figure 3d). It is known that this turn−like structure stabilizes the β−turn−β structure and promotes the formation of the folding core for Aβ fibrillation [36,60]. From the content of secondary structures at equilibrium, it was found that the β−bridge structure (0.5%) was still present in the Aβ_40_−PSMα3o system (Appendix A), which was formed in the Phe4−Ser8 region of Aβ_40_ (Appendix A), indicating that the PSMα3o could not inhibit the formation of the β−sheet structures in Aβ_40_. This is consistent with the results of the ThT fluorescence and CD experiments (Figure 2a and Table 1), as the PSMα3o did not significantly affect the maximum ThT fluorescence intensity or the content of the β−sheet structure of Aβ_40_, which also suggested that the PSMα3o weakly affected the conformational transitions of the Aβ_40_ monomer. Furthermore, the MM−PBSA results showed that the binding energy of the PSMα3o to the Aβ_40_ monomer was −135.5 kJ/mol, significantly lower than the PSMα3m to the Aβ_40_ monomer (−270.7 kJ/mol) (Appendix A). This is mainly due to the significantly weaker van der Waals hydrophobic energy between Aβ_40_ and the PSMα3o (−251.0 kJ/mol) than the PSMα3m (−373.5 kJ/mol). Free energy decomposition showed that the PSMα3o mainly combined with the central hydrophobic core (CHC) and the C−terminal region of Aβ_40_ (His13−Val36) (Figure 6a), which plays a decisive role in the Aβ oligomerization [50]. It can be found that some hydrophobic residues (Leu7, Phe10, and Phe18) of the PSMα3o contributed significantly to the high binding energy, although their contribution was generally lower than that of the PSMα3m (Figure 6b). In addition, only one stable H−bond was observed in the Aβ_40_−PSMα3o complex (Appendix A), indicating that the H−bonding between the Aβ_40_ monomer and the PSMα3o was much weaker than for the PSMα3m. These results suggest that the weaker H−bonding and hydrophobic interactions resulted in a lower binding energy of the PSMα3o to the Aβ_40_ monomer than that of the PSMα3m, leading to the weak effect of the PSMα3o on the conformational transitions of Aβ_40_.

According to the above studies, it may be concluded that electrostatic and hydrophobic interactions contributed to the binding of the PSMα3m to the Aβ_40_ monomer. PSMα3m had a high binding affinity for the Aβ_40_ monomer, with a total of six H−bonds formed between the two monomers, and some hydrophobic residues (Met1, Phe3, Leu7, Leu14, and Phe18) of the PSMα3m provided hydrophobic interactions with Aβ_40_. The hydrophobic interactions and H−bonding led to the tight binding of the PSMα3m to the Aβ_40_ monomer and inhibited the conformational transitions of Aβ_40_ from the α−helix to β−sheet structure. In contrast, the PSMα3o only formed one stable H−bond with the Aβ_40_ monomer, and the weaker H−bonding and hydrophobic interactions resulted in a lower binding free energy of the PSMα3o to the Aβ_40_ monomer, leading to the small role of the PSMα3o in inhibiting the formation of the β−sheet structure in Aβ_40_.

## 4. Conclusions

We have explored the effect of PSMα3, a bacterial amyloid with a unique cross−α structure, on the aggregation kinetics, molecular structures, and conformational transitions of the Aβ_40_ monomer using experimental and computational approaches. The results showed that the PSMα3 monomer inhibits Aβ_40_ aggregation in a concentration−dependent manner, prolongs its aggregation lag phase, and redirects the aggregation pathway of Aβ_40_ to form granular structures. In contrast, the PSMα3 oligomer promotes the generation of the β−sheet structure and shortens the lag phase of Aβ_40_ aggregation. The cross−α structure of PSMα3 plays an important role in the aggregation of Aβ_40_. The higher the cross−α content of PSMα3, the more obvious the promotion effect on the aggregation of Aβ_40_. MD simulations further support these observations and identify the key regions and amino acid residues of Aβ_40_ and PSMα3 in the interactions. The PSMα3 monomer binds to the Asp1−Ala2 and His13−Val36 regions in the Aβ_40_ monomer via hydrophobic and electrostatic interactions, which prevents the conformational transitions of Aβ_40_ from the initial α−helix structure to the β−sheet structure. In contrast, the PSMα3 oligomer mainly binds to the CHC and the C−terminal region of the Aβ_40_ monomer through weak H−bonding and hydrophobic interactions, resulting in a weak role in the conformational transitions of Aβ_40_. This work reveals different interactions between PSMα3 and Aβ_40_, providing deeper insights into the complex interactions between bacterial amyloids and AD−associated pathogenic Aβ.

Since the formation of cross−α fibrils by PSMα3 enhances the toxicity to human cells [18] and promotes the formation of the β−sheet structures of Aβ_40_, subsequent studies should direct towards examining the co−aggregation of PSMα3 and Aβ_40_ in vivo. In addition, it has been shown that some amyloid−derived fragments with cross−seeding capacity can bind to amyloids and inhibit its aggregation [61,62]. The PSMα3 monomer has a strong affinity for Aβ_40_ and can significantly inhibit the formation of the β−sheet structure, and its derived fragments may serve as potential inhibitors of Aβ aggregation in vivo. The new findings may provide molecular insights into a potential association between AD and the intestinal flora and provide a potential strategy for the design of amyloids inhibitors based on cross−seeding.

## Figures and Tables

**Figure 1 biomimetics-08-00459-f001:**
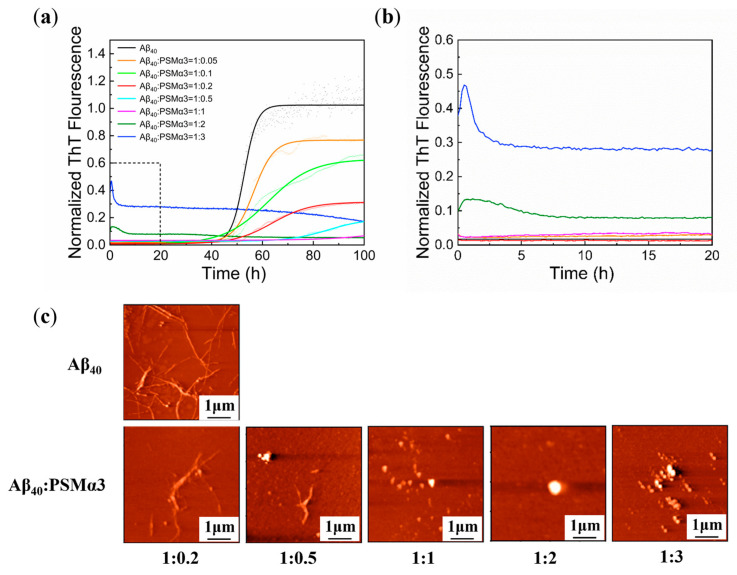
Effect of the PSMα3 monomer on Aβ_40_ aggregation: (**a**) aggregation kinetics of Aβ_40_ (25 μM) cultured with or without the PSMα3 monomer at different concentrations measured by ThT fluorescence assay; (**b**) the enlarged version of the dotted frame area in (**a**); (**c**) AFM images of Aβ_40_ aggregates cultured with or without the PSMα3 monomer of different concentrations at 37 °C for 100 h.

**Figure 2 biomimetics-08-00459-f002:**
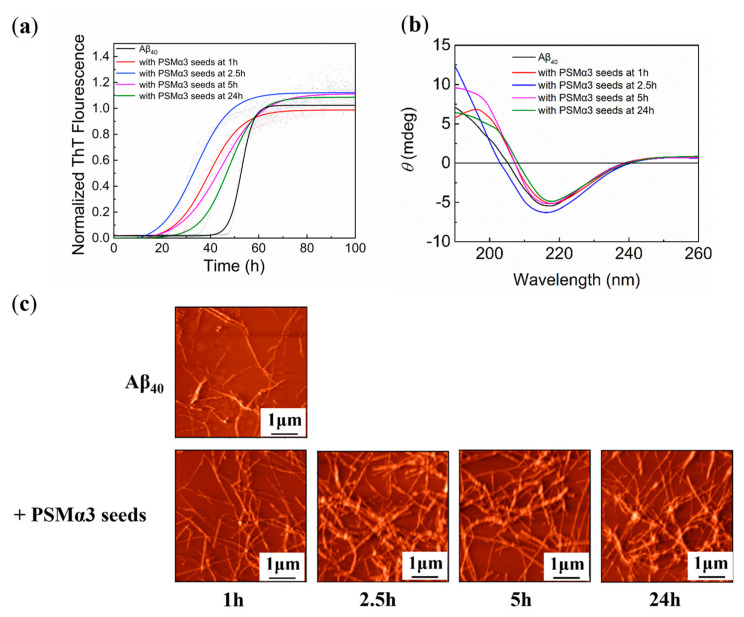
Effect of PSMα3 seeds on Aβ_40_ aggregation: (**a**) aggregation kinetics of Aβ_40_ (25 μM) cultured without or with PSMα3 seeds (5 μM) at 1 h, 2.5 h, 5 h, and 24 h measured by ThT fluorescence assays for 100 h; (**b**) CD spectra of Aβ_40_ aggregates cultured with or without PSMα3 seeds at 1 h, 2.5 h, 5 h, and 24 h for 100 h; (**c**) AFM images of Aβ_40_ cultured with or without PSMα3 seeds at 37 °C for 100 h.

**Figure 3 biomimetics-08-00459-f003:**
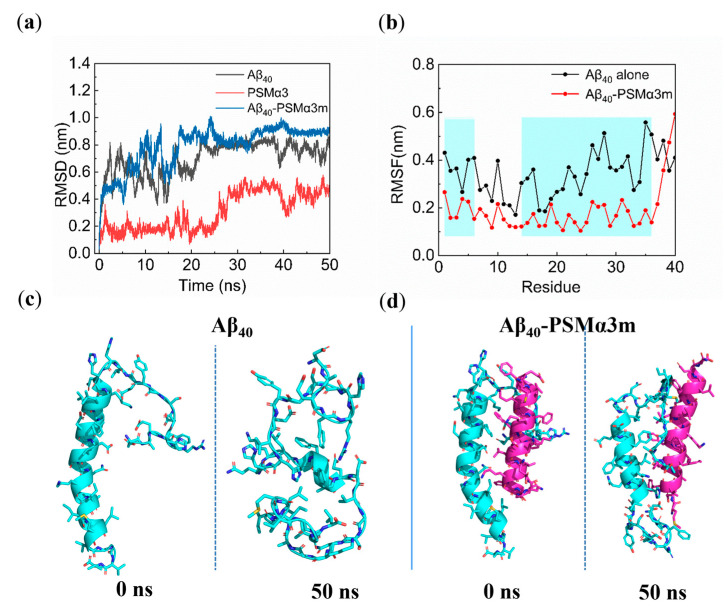
MD simulations of the PSMα3 monomer (PSMα3m) and the Aβ_40_ monomer and their complex: (**a**) RMSD of Aβ_40_, PSMα3, and Aβ_40_−PSMα3m complex; (**b**) RMSF of each residue of Aβ_40_ without or with PSMα3m; (**c**,**d**) initial (0 ns) and equilibrium (50 ns) states of (**c**) Aβ_40_ and (**d**) Aβ_40_−PSMα3m complex. Blue and purple represent the Aβ_40_ monomer and PSMα3m, respectively.

**Figure 4 biomimetics-08-00459-f004:**
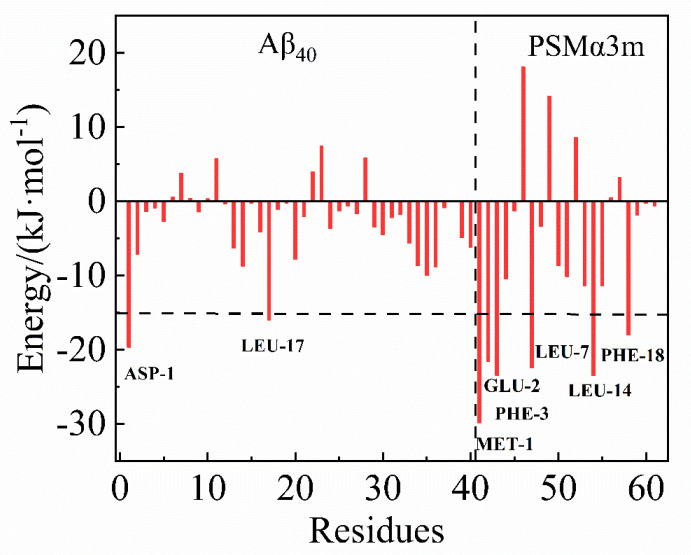
Energy decomposition diagram of each amino acid residue in the Aβ_40_−PSMα3m complex.

**Figure 5 biomimetics-08-00459-f005:**
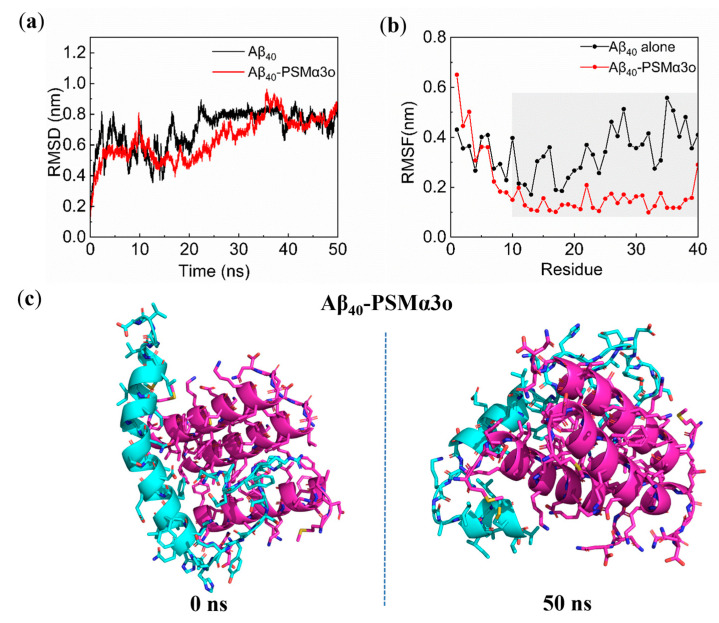
MD simulations of Aβ_40_−PSMα3 oligomer (PSMα3o) complex: (**a**) RMSD of Aβ_40_ and Aβ_40_−PSMα3o system; (**b**) RMSF of each amino acid residue of Aβ_40_ without or with PSMα3o; (**c**) initial (0 ns) and equilibrium (50 ns) states of the Aβ_40_−PSMα3o system. Blue and purple represent the Aβ_40_ monomer and PSMα3o, respectively.

**Figure 6 biomimetics-08-00459-f006:**
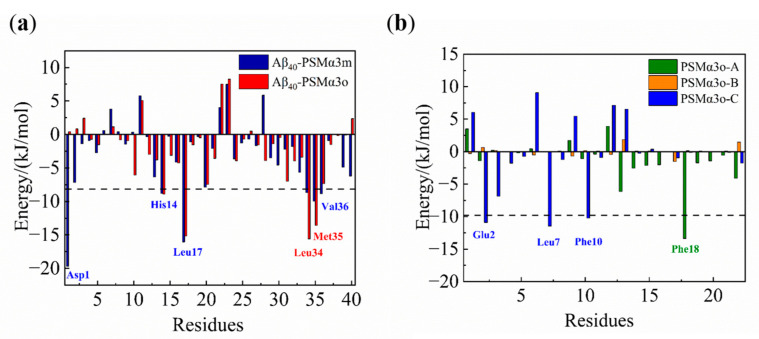
Energy decomposition diagram of each residue of (**a**) Aβ_40_ and (**b**) PSMα3o. Blue and red represent residues of Aβ_40_ in the Aβ_40_−PSMα3m and Aβ_40_−PSMα3o complexes, respectively. Green, yellow and blue separately represent the three chains of PSMα3o.

**Table 1 biomimetics-08-00459-t001:** Secondary structure distributions of Aβ_40_ aggregates incubated with PSMα3 seeds of different culture times calculated from the CD spectrum.

Sample (%)	α−Helix	β−Sheet	Turn	Others
Aβ_40_	10	74.4	3.8	11.8
with PSMα3 seeds at 1 h	7.8	73.2	1.2	17.9
with PSMα3 seeds at 2.5 h	29.3	63.6	7.1	0
with PSMα3 seeds at 5 h	14.2	81.9	3.8	0
with PSMα3 seeds at 24 h	18.3	79.1	2.6	0

## Data Availability

Not applicable.

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
