# Peer review of "Effect of Bacterial Amyloid Protein Phenol−Soluble Modulin Alpha 3 on the Aggregation of Amyloid Beta Protein Associated with Alzheimer’s Disease"

_biomimetics, 2023, doi:10.3390/biomimetics8060459_

Round 1
Reviewer 1 Report
The authors report the study of the effect of Bacterial Amyloid Protein PSMα3 on the aggregation of Aβ40. The results indicate that PSMα3 monomer prevents Aβ40 from transforming to β-sheet structure to inhibit the aggregation, while PSMα3 seeds still perform cross-seeding acceleration. These findings are interesting and shed lights on the understanding on the relationship between intestinal flora and pathogenic amyloid and neurodegenerative diseases. In general, the experimental data and MD simulations are consistent. The major conclusions are supported by the results. This manuscript is well-written. However, there are concerns to be addressed to improve the quality before the consideration of publication.
Comments:
In the first part, the authors characterized the secondary structures and morphologies of Aβ40 and PSMα3 aggregates and the results mostly are consistent with previous observations. However, the AFM images of PSMα3 show granular but not the fibril like structure as shown in ref 19 (Fig. 1 & Fig, S1). Why does the CD spectrum of PSMα3 aggregates not showing the two negative peaks at 208 and 222 nm? Does this indicate that these aggregates are not cross-α structured? Please explain these inconsistencies. Also, in line 161, the cited ref 31 reported different cross-b structures of PSMα1 and PSMα4, which is different from PSMα3.
When mixed with Aβ40 and during the aggregation reaction, will the PSMα3 still keep monomeric given that it tends to form self-assembled aggregates?
In line 190 and 219, the authors cited ref 35 and 36 to support the conclusion that Aβ40 and PSMα3 form heterogenous oligomer, which in my opinion may be prejudiced, since there is no direct experimental evidence for this statement.
In line 227, it is still premature to claim that it is the “secondary nucleation” that is accelerated based on current result. To clarify this, a kinetic analysis (DOI: 10.1038/nprot.2016.010) to determine the kinetic parameters of secondary nucleation rate will give clear evidence.
Reviewer 2 Report
This article investigates the ability of a bacterial amyloid to modulate the aggregation of Ab40, which is important to AD. It is interesting and will be of interest to the aggregation community.
Major point: A weakness that could be easily addressed is an over-reliance on ThT without independent controls. I strongly recommend the authors carry out dot-blot assays to confirm the ability of PSMa3 monomers and oligomers to inhibit and accelerate Ab40 aggregation, respectively. Ideally, they would use a sequence specific antibody like 6E10 to confirm the depletion of soluble Ab40 from the supernatant over the aggregation reaction, the OC antibody to confirm the formation of fibrillar aggregates in the pellet, and an antibody against PSMa3 if one exists. This would give much more confidence to their conclusions.
Minor points:
It would have been nice if the authors assessed lower concentrations of PSMa3 in the Ab40 kinetic experiments, too, as the effect in the PSMa3 monomer experiments are large at the lowest concentration of the bacterial amyloid.
It would also be extremely interesting for the authors to carry out experiments with fibrillar Ab40 seeds, and to use the unseeded and seeded data to understand which microscopic steps are inhibited or sped up by the various forms of PSMa3. A program like AmyloFit would help them do this.
Table S1: what do the error represent? How many experiments/replicates? Please make sure the number of replicates and independent experiments are clearly stated for all experiments.
Reviewer 3 Report
Phenol soluble modulins (PSM) α3, as the most virulent proteins secreted by Staphylococcus aureus, has attracted much attention. In this work, Sun group studied the effect of PSMα3 with unique cross-α fibril architecture on the aggregation of pathogenic Aβ40 of AD via the extensive biophysical characterizations. The results proposed that PSMα3 monomer inhibited the aggregation of Aβ40 in a concentration-dependent manner and changed the aggregation pathway to form granular aggregates. However, PSMα3 oligomers promoted the generation of β-sheet structure, thus shortening the lag phase of Aβ40 aggregation. Moreover, the higher the cross-α content of PSMα3, the stronger effect of the promotion, indicating that the cross-α structure of PSMα3 plays a crucial role in the aggregation of Aβ40. Further molecular dynamics (MD) simulations have shown that the Met1-Gly20 region in the PSMα3 monomer can be combined with the Asp1-Ala2 and His13-Val36 regions in Aβ40 monomer by hydrophobic and electrostatic interactions, which prevents the conformational conversion of Aβ40 from α-helix to β-sheet structure. The research has unraveled molecular interactions between Aβ40 and PSMα3 of different structures and provided a deeper understanding of the complex interactions between bacterial amyloid protein and AD-related pathogenic Aβ. Overall, this work (both fundamental research and simulation results) is quite comprehensive and well organized. I suggest to publish it after minor revision.
1. Basically, the 22-residue peptide phenol-soluble modulin α3 (PSMα3) is the most cytotoxic and lytic member of the PSMs family. Will it further affect the (i) inhibition effect of the aggregation of Aβ40 or (ii) promotion the generation of β-sheet structure by their oligomers in vivo? If so, I think authors should add these discussions in the conclusion or results and discussion. After all, we will not stop by the in vitro studies for the following disease detection/treatment.
2. Figure 1. The authors did not give very thorough discussions on the ratio effect. Typically, readers would like to know the details compared to the Figures. More result details should be added here.
3. For simulations. Is 50 ns really the time for the equilibrium status of the complex?
Please proofread it before submitting the revisions.
Reviewer 4 Report

The language is fine, only minor editing in grammar is needed.
Round 2
Reviewer 1 Report
I'm satisfied with the authors' response and no more concerns to be addressed.
Author Response
We sincerely appreciate your valuable comments and suggestions for this paper.
Reviewer 2 Report
The statement "It indicated that the PSMα3 monomer mainly inhibited the secondary nucleation pathway of Ab40 aggregation" cannot be made without doing the seeded aggregation experiments (i.e. using AB40 fibrils to bypass primary nucleation at low seed concentrations and both primary and secondary nucleation at high see concentrations; see Habchi PNAS 2017 for an example of this with Ab42). This statement should be corrected so the data are consistent with their conclusions. Same for lines 244-249, you need to do the experiments to make the claims. I recommend editing the text at this stage instead of doing more kinetic experiments.
In response to "Due to the lack of experimental materials and limited funding, we are unable to carry out this experiment in a short period of time. We hope that in the future, when conditions permit, it will be possible to conduct relevant studies to draw more interesting conclusions." the OC and 6E10 or WO2 antibodies are widely available to recognize fibrillar and monomeric Ab40, respectively. These are straightforward measurements that are important controls for your study to exclude ThT-based artifacts, which forms the basis of all your conclusions with the AFM and MD.
"The experiment with Aβ40 seeds has a large experimental error and could not be completed in a short time" seeding experiments are very straightforward. Grow fibrillar Ab40, add the fibrils to the monomer, and go. I appreciate if you want to omit these measurements and address my first point above, but be aware these experiments are very doable.
